# DOT1L-mediated murine neuronal differentiation associates with H3K79me2 accumulation and preserves SOX2-enhancer accessibility

Francesco Ferrari[1,2], Laura Arrigoni [1], Henriette Franz [3], Annalisa Izzo[3], Ludmila Butenko[3], Eirini Trompouki [1], Tanja Vogel [3,4✉] & Thomas Manke [1✉]

During neuronal differentiation, the transcriptional profile and the epigenetic context of neural committed cells is subject to significant rearrangements, but a systematic quantification of global histone modification changes is still missing. Here, we show that H3K79me2 increases and H3K27ac decreases globally during in-vitro neuronal differentiation of murine embryonic stem cells. DOT1L mediates all three degrees of methylation of H3K79 and its enzymatic activity is critical to modulate cellular differentiation and reprogramming. In this context, we find that inhibition of DOT1L in neural progenitor cells biases the transcriptional state towards neuronal differentiation, resulting in transcriptional upregulation of genes marked with H3K27me3 on the promoter region. We further show that DOT1L inhibition affects accessibility of SOX2-bound enhancers and impairs SOX2 binding in neural progenitors. Our work provides evidence that DOT1L activity gates differentiation of progenitors by allowing SOX2-dependent transcription of stemness programs.

[1] Max Planck Institute of Immunobiology and Epigenetics, Freiburg, Germany. [2] Faculty of Biology, University of Freiburg, Freiburg, Germany. [3] Institute of Anatomy and Cell Biology, Department of Molecular Embryology, Faculty of Medicine, University of Freiburg, Freiburg, Germany. [4] Center for Basics in NeuroModulation (NeuroModul Basics), Faculty of Medicine, University of Freiburg, Freiburg, Germany. ✉email: tanja.vogel@anat.uni-freiburg.de; manke@ie-freiburg.mpg.de

During differentiation, eukaryotic cells undergo large changes affecting their structural, functional and metabolic profiles. The process is accompanied by major rearrangements of the epigenetic and transcriptional profile, which are driven by the synergistic effects of epigenetic enzymes and transcription factors[1].

Epigenetic and transcriptional changes driving neuronal differentiation have been characterized[2,3], but few efforts have been directed towards a comprehensive description of global histone modification dynamics of neural committed cells[4]. Previous investigations were limited by the use of semi-quantitative and low-throughput methods (e.g., immunoblotting and imaging). Recent developments in quantitative chromatin immunoprecipitation followed by sequencing (ChIP-seq) have overcome these technical limitations and they now allow detection of genome-wide global changes in histone modifications across conditions in a high-throughput manner[5–7].

Various epigenetic enzymes are important for the orchestration of neuronal differentiation[2,8]. Among these, Disruptor of Telomeric silencing 1 Like (DOT1L) has been recently identified as a critical player in the differentiation process[9–12]. DOT1L is a highly conserved histone methyltransferase that catalyzes the mono-, di- and trimethylation of lysine 79 of histone H3 (H3K79me1/me2/me3)[13]. Since its first characterization in yeast as a disruptor of telomeric silencing upon gain or loss of function[14,15], the protein has been recognized to be involved in a variety of biological processes, such as cell cycle control[16], DNA repair[13], gene expression[17], differentiation, and reprogramming[18]. DOT1L regulates cardiomyocyte differentiation and maturation[19,20] and chondrocyte differentiation[21], while the modulation of its enzymatic activity was shown to be critical for cellular reprogramming efficiency[9]. Within the neural lineage, DOT1L prevents premature cell cycle exit and depletion of the neural progenitor pool and it is necessary for proper neuronal differentiation[12,22,23].

DOT1L plays a prominent role in certain forms of leukemia. Interestingly, some studies in this field identified specific perturbations of the chromatin context that manifest upon blocking of DOT1L and indicate crosstalk between H3K79me2 and histone acetylation. Chen et al. show that *Dot1l* knock-down results in the establishment of repressive chromatin states around Myeloid/ Lymphoid Or Mixed-Lineage Leukemia (MLL) target genes. This evidence suggests that the presence of H3K79 methylation may be critical to prevent deacetylation through the activity of SIRT1-complexes[24]. Loss of DOT1L activity also results in decreased acetylation and reduced frequency of promoter-enhancer interactions at H3K79me2-marked enhancers[25]. Currently, it is not clear whether the molecular perturbations described in leukemia are relevant for the differentiation phenotypes described in other model systems, and whether DOT1L activity targets enhancers in physiological developmental settings.

In this work, we use mouse embryonic stem cells (mESC) and their in vitro derived neural progeny (NPC48h) to systematically characterize the global dynamics of the epigenetic landscape during neuronal differentiation[26]. For both mESC and NPC48h, we further investigate whether competitive inhibition of DOT1L with Pinometostat (EPZ5676) affects the establishment of chromatin states and cell-type-specific transcriptional programs.

We show that the global levels of H3K79me2 increase genome-wide during the differentiation process. Our data indicate that DOT1L inactivation causes the onset of a transcriptional program which primes mESC-derived NPC towards neuronal differentiation. We further show that acute DOT1L inhibition is associated with reduced accessibility of intronic and intergenic enhancers that are bound in vivo by the stemness-conferring transcription factor SOX2.

## Results

**NGS dataset of murine in vitro neuronal differentiation.** To characterize the epigenetic and transcriptional changes during neuronal differentiation and to study the cell-type-specific causal contribution of DOT1L to the neuronal differentiation process, we generated and integrated a multi-omics dataset encompassing comprehensive epigenomes of seven histone modifications (H3K4me1, H3K4me3, H3K9me3, H3K27ac, H3K27me3, H3K36me3, and H3K79me2), transcriptomes and chromatin accessibility profiles of mESC and NPC48h treated with dimethyl sulfoxide (DMSO) or Pinometostat (EPZ5676, EPZ). To assess quantitative epigenetic changes, we used RELACS, a chromatin barcoding strategy for multiplexed and quantitative ChIP-seq[5] (Fig. 1a).

We first assessed the biological coherence of the generated multi-omics datasets during differentiation in the control condition. As expected, the transcriptome segregated mESC and NPC48h into two distinct groups (Fig. 1b, upper panel). A clear separation between mESC and NPC48h was also obtained from dimensionality reduction of the epigenome (Fig. 1b, lower panel). The chromatin-based separation between cell types was most strongly determined by active histone modifications (Supplementary Fig. 1a). Differential gene expression analysis showed dynamic genes [abs(log2 fold-change) > 1, adjusted $p$-value < 0.01] to be prevalently upregulated in NPC48h compared to mESC (Supplementary Fig. 1b). Consistently, protein-coding genes showed higher coverage of the co-transcriptionally regulated marks H3K79me2 and H3K36me3 on the 5′end and 3′end of the gene body respectively, in NPC48h compared to mESC (Supplementary Fig. 1c). GO term enrichment analysis of differentially expressed genes (DEG) between NPC48h and mESC showed a neuronal signature in the upregulated set, providing evidence for the neuronal transcriptional identity of the differentiated cells (Supplementary Fig. 1d).

Next, we modeled the relationship between transcriptional dynamics and changes in histone modifications around transcriptional start sites (TSS) and transcriptional termination sites (TTS). (H3K4me3, H3K27ac, H3K4me1: TSS ± 1 kb; H3K79me2, H3K27me3, H3K9me3: TSS−1 kb +3 kb; H3K36me3: TTS−3 kb). As expected, chromatin changes correlated with transcriptional changes (Supplementary Fig. 1e), but the epigenetic features were collinear and thus redundant. To decrease redundancy, we ranked histone PTM dynamics based on their relevance for predicting transcriptional changes by fitting a regularized linear model to our dataset. We found that H3K27ac, H3K36me3, and H3K79me2 were the most relevant predictive features for transcriptional dynamics upon neural differentiation, followed by an interaction term between H3K79me2 and H3K27ac (H3K79me2:H3K27ac) (Supplementary Fig. 1f). Our model (model 1, m_1) successfully captured the observed expression trends ($R^2 = 0.44$) (Supplementary Fig. 1g, left panel) and resulted in a better fit compared to previous attempts with more complex models[27].

The inclusion of the interaction term did not lead to overfitting and increased the accuracy of log2 fold-change prediction for genes that were strongly upregulated during the differentiation process. Interestingly, many of the genes that were mostly affected by the interaction term were known targets of retinoic acid (RA) mediated in vitro neuronal differentiation (e.g., *Hoxa* and *Hoxb* clusters, *Ascl1, Zic1, Zic4, Pou3f2, Pou3f3, Nhlh2, Lhx1*)[28,29] (Supplementary Fig. 1g, right panel).

Together, these data provide evidence for the coherence of the generated multi-omics dataset and show that in vitro neuronal differentiation of mESC correlates with relative epigenetic and transcriptional activation. We show that gene expression changes in our model can be predicted using a linear combination of a

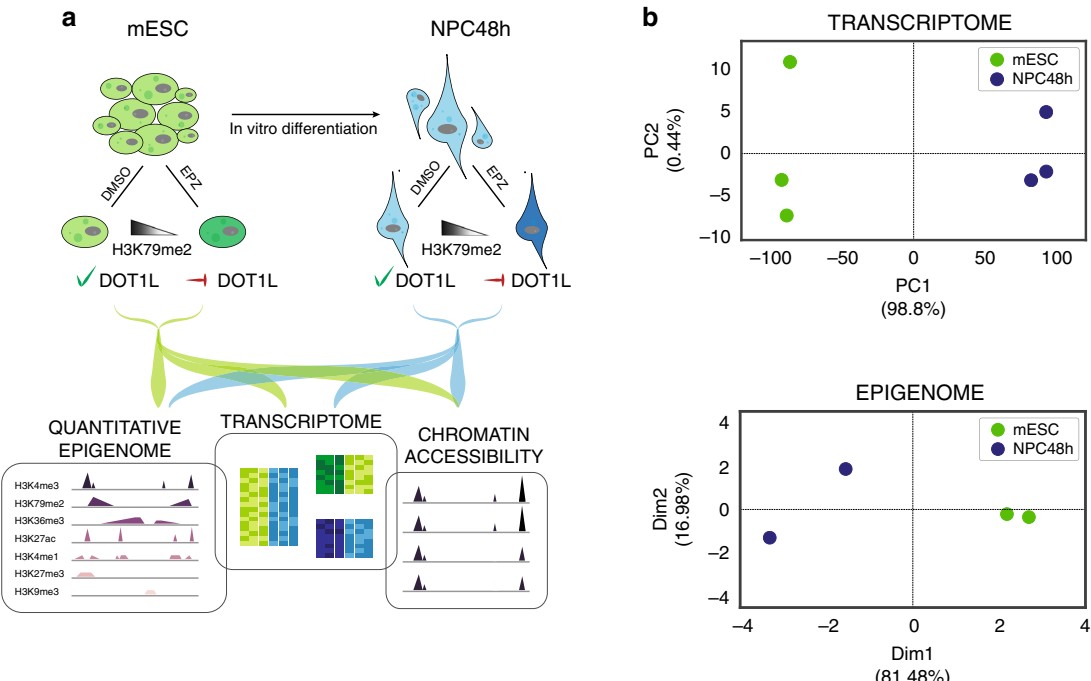

**Fig. 1 Experimental design and QC of multi-omics datasets of murine in vitro neuronal differentiation. a** Experimental design of this study. We differentiate mESC towards neural progenitors (NPC48h) in vitro. We treat mESC and NPC for 48 h either with the DOT1L inhibitor EPZ5676 (10 μM) or with DMSO (1/1000) as control. For each sample, we generate transcriptomics data via RNA-seq and comprehensive epigenomes using quantitative ChIP-seq. We further map the accessible chromatin landscape for NPC48h using ATAC-seq. **b** Upper panel: principal component analysis of the transcriptome of mESC and NPC48h on the top 500 most variable genes (rlog transformed counts) shows a separation between the two cell types on the first principal component. Lower panel: multiple factor analysis of the epigenome of mESC and NPC48h computed over the top 500 most variable 2 kb windows for each histone modification yield similar results. Biological replicates are denoted by the same color.

subset of histone modification changes (H3K27ac, H3K36me3, H3K79me2) and that the interaction between H3K27ac and H3K79me2 is relevant to predict expression changes of genes driving neuronal differentiation.

**H3K27ac and H3K79me2 undergo opposite global changes.** The computation of histone modification changes in a classic ChIP-seq experiment imposes a per-sample normalization that prevents the detection of global shifts. In contrast, the RELACS barcoding strategy we employed in this work allowed for quantitative estimations of genome-wide global epigenetic changes between samples. To assess the global dynamics of each histone PTM during the differentiation process, we estimated global scaling factors from sequencing data of control samples by computing pairwise ratios of input normalized read counts allocated to each sample after demultiplexing[5].

We observed that H3K4me3, H3K4me1, and H3K27me3 levels in NPC48h did not deviate globally from mESC reference levels. H3K36me3 and H3K9me3 showed a mild global increase. Strong global changes were instead observed for H3K27ac and H3K79me2 during neuronal differentiation, with the former decreasing by a factor of ~2 (2.3 ± 0.12) and the latter increasing by a factor of ~4 (3.9 ± 0.05) (Fig. 2a).

Global shifts of histone PTM levels can be caused by two possible mechanisms. (1) Histone marks can accumulate on specific loci, resulting in local enrichment compared to flanking regions (so called "peaks"). A global shift can occur if the number and magnitude of histone PTM local enrichment changes across conditions. In this work, we refer to this mechanism as a locally driven global change (Fig. 2b, left panel). (2) Alternatively, histone PTM can accumulate or be removed homogeneously over the whole genome, causing a base-level global gain/loss of the

signal. Traditional ChIP-seq methods are unable to detect these global shifts. In this work, we refer to this mechanism as a genome-wide driven global change (Fig. 2b, right panel). Furthermore, global changes may result from a combination of (1) and (2).

To understand if the measured global changes were genome-wide or locally driven, we visualized locus-specific changes of H3K4me3, H3K27ac, H3K36me3, and H3K79me2 levels between NPC48h and mESC, on annotated genomic features (H3K4me3 and H3K27ac: TSS ± 2 kb; H3K79me2: TSS + 3 kb; H3K36me3: TTS−3 kb). Results indicated that H3K4me3 levels were unaffected upon differentiation in both background and locally enriched regions. H3K36me3 levels did not change globally in background regions but showed a mild increase in locally enriched regions. This indicates that H3K36me3 global change was mostly locally driven. In contrast, loss of H3K27ac and gain of H3K79me2 affected background and locally enriched loci to an almost equal extent. This observation indicates that the global changes measured for these two marks were genome-wide driven (Fig. 2c).

In summary, our analyses show that H3K27ac and H3K79me2 levels change globally during in vitro neuronal differentiation, with opposite trends and through genome-wide acting mechanisms.

**H3K79me local changes correlate with transcriptional changes.** It has been reported that H3K79me2 local enrichment is the best linear predictor of gene expression levels[30], but the functional relevance of the global H3K79me2 increase during neuronal differentiation remains to be clarified, particularly for transcription. Therefore, we asked whether differential H3K79me2 local

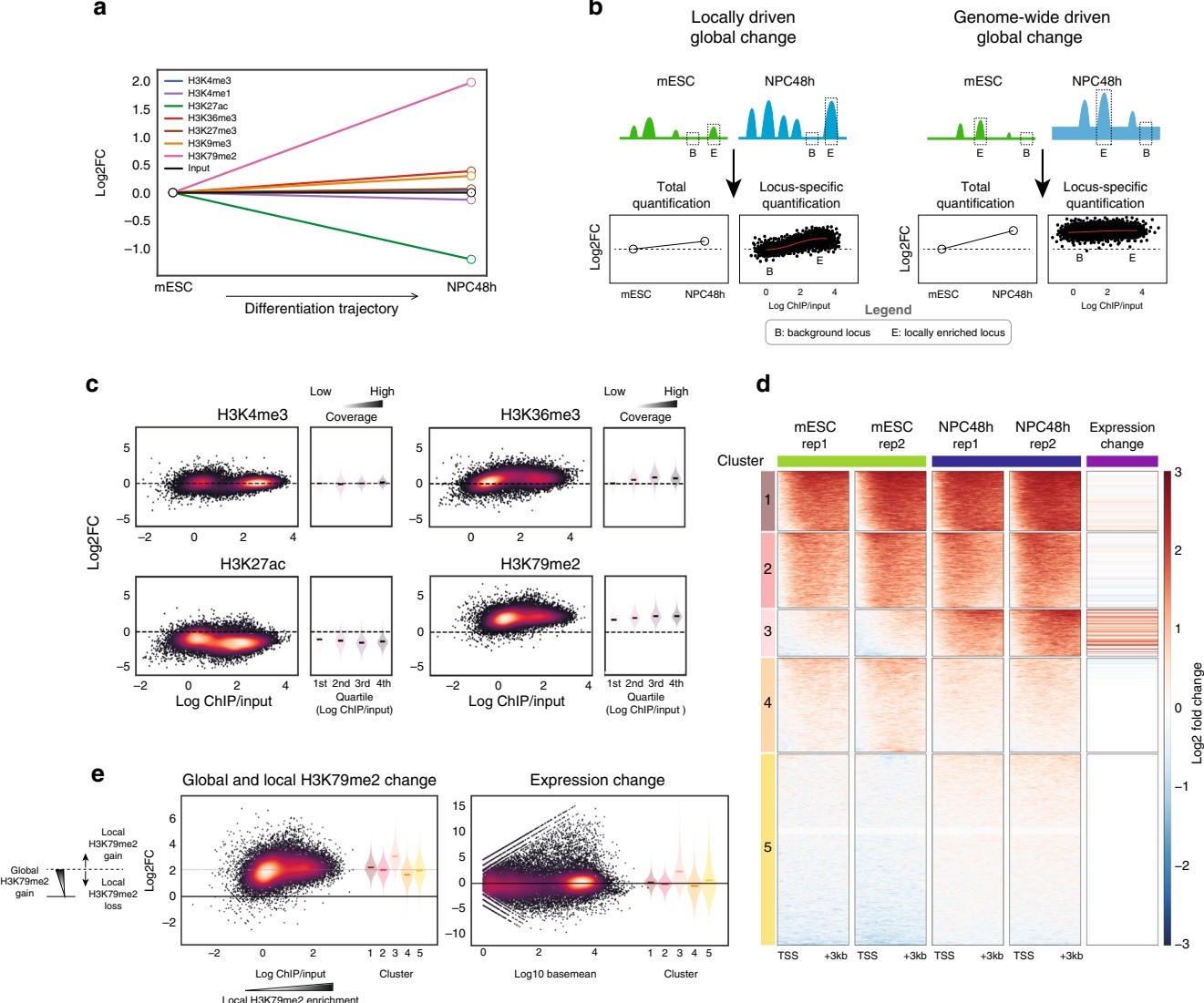

**Fig. 2 H3K27ac and H3K79me2 undergo opposite global changes independent of transcriptional changes during in vitro neuronal differentiation. a** Representation of the global scaling factors (log2 transformed) estimated in NPC48h using quantitative ChIP-seq with respect to the reference mESC level, for the seven histone modifications included in this study ($N = 2$). **b** Model to illustrate that global histone modification changes can result from two different scenarios. Left panel: locally driven global changes may follow from increased number and/or magnitude of histone PTM local enrichment. Right panel: genome-wide driven global changes may follow from a genome-wide accumulation of a mark on both locally enriched regions ("peaks") and on background regions. **c** MA plots showing the mean coverage ($x$-axis) and log2 fold-change ($y$-axis) of four histone marks computed on bins overlapping annotated genomic features for the contrast NPC48h vs mESC (H3K4me3 and H3K27ac: transcription start site (TSS) ± 2 kb; H3K79me2: TSS + 3 kb; H3K36me3: transcription termination site (TTS)−3 kb). Next to each MA plot, we show a summary plot showing the global fold-change distribution ($y$-axis) for each quartile of mean coverage ($x$-axis). **d** k-mean clustering ($k = 5$) of H3K79me2 enrichment 3 kb downstream of TSS of protein-coding genes. A standard scaling method (by sequencing depth) and normalization (by input-control) was used. This highlights changes with respect to local background. The last column shows the log2 fold-changes in the expression of the respective genes in each cluster for the contrast NPC48h vs mESC. Genes gaining local H3K79me2 enrichment tend to be upregulated during in vitro neuronal differentiation. **e** Left panel: MA plot showing the mean coverage ($x$-axis) and the global change in H3K79me2 ($y$-axis) computed on a window 3 kb downstream of TSS of protein-coding genes, next to a violin plot showing the global H3K79me2 changes for each of the five clusters previously identified in (**d**). The scheme on the right helps the interpretation of the global and local H3K79me2 changes. Right panel: MA plot of gene expression changes for the contrast NPC48h vs mESC, next to a violin plot showing the expression changes of genes clustered according to (**d**).

enrichment, and/or the global H3K79me2 increase, associated with transcriptional dynamics.

To address this question, we stratified protein-coding genes in five clusters using standard log2-ratio scores between sequencing-depth normalized H3K79me2 and input, individually for each cell type. It should be noted that this approach corresponded to a traditional normalization absorbing all global changes. We quantified scores on a 3 kb window downstream of TSS of mESC

and NPC48h (Fig. 2d). We observed that genes included in clusters 1 and 2 were locally enriched in both cell types and despite the global gain in H3K79me2, their expression levels did not change during differentiation (Fig. 2e). Cluster 3, on the other hand, identified a group of genes that gained H3K79me2 locally in addition to the global increase during development. These genes presented a clear axonogenic signature (Supplementary Fig. 2a), and their expression level significantly increased during

differentiation (Fig. 2d, e). A mild reduction of H3K79me2 local enrichment was detected on genes belonging to cluster 4, but no major effect was observed at the transcriptional level. Cluster 5 identified genes with neither H3K79me2 enrichment nor dynamic expression.

Together, these data show that, in our system, dynamic local enrichment of H3K79me2 associates with transcriptional activation of genes critical for neuronal development. Global accumulation of H3K79me2 did not associate with transcriptional changes.

**DOT1L inactivation biases the transcriptome of NPC48h**. In yeast, fly, mouse, and human, H3K79me2 is generally associated with transcriptional activity. The mark localizes to euchromatic regions and its enrichment strongly correlates with gene expression level[13]. Yet, little is known about the causal relevance of DOT1L methyltransferase activity for the transcriptional process. To investigate the cell-type-specific causal contribution of DOT1L enzymatic function for the genome-wide transcriptional activity and for the overall epigenetic context, we inhibited the enzyme in mESC and NPC by treating cells for 48 h with the S-adenosyl methionine (SAM) competitor Pinometostat (EPZ5676, EPZ). Subsequently, we quantified transcriptional and epigenetic changes compared to cells treated with DMSO as control.

EPZ treatment successfully inhibited DOT1L methyltransferase activity, as shown by the reduction of all three degrees of H3K79 methylation (Supplementary Fig. 3a). Quantification after immunoblotting showed that the total H3K79me2 signal equals 47.8% ± 5.7% and 59.6% ± 4.7% of the reference DMSO level in mESC and NPC48h respectively (Fig. 3a). Quantification based on RELACS ChIP-seq confirmed this trend and indicated that the total H3K79me2 signal was equal to 44.9% ± 2.4% and 64.2% ± 5.8% of the reference DMSO level in mESC and NPC48h respectively (Fig. 3a). To test if we could resolve signal loss at single-locus resolution, we computed locus-specific changes of H3K79me2 over the previously defined five clusters (see Fig. 2d). Results indicated that signal loss can be read as a function of H3K79me2 local enrichment (Fig. 3b), where weakly marked loci (cluster 5) lost comparably less H3K79me2 signal than strongly marked loci (cluster 1).

To study the effects of DOT1L inhibition on the transcriptome, we first identified DEG across treatment groups. EPZ treatment caused a mild alteration of the transcriptome in both mESC and NPC48h, as indicated by principal component analysis (PCA) and sample clustering on normalized count data (Supplementary Fig. 3b), where the main variability came from biological replicates rather than treatment.

DEG showed only moderate log2 fold-changes (Supplementary Fig. 4a). The transcriptional alteration was more pronounced in mESC than NPC48h. 58 genes were commonly differentially expressed in both cell types (adjusted *p*-value < 0.05) (Supplementary Fig. 4b). Transcriptional deregulation in NPC48h was not associated with gain of H3K79me2 during differentiation (φ coefficient = 0.046, 65 DEG of 2170 genes gaining H3K79me2 during neuronal differentiation) (Supplementary Fig. 4c).

For mESC, gene set enrichment analysis (GSEA) identified significant pathways sharing an immunological and stress-induced pro-apoptotic molecular signature (Supplementary Fig. 4d, left panel). For NPC48h, GSEA showed the deregulation of Wnt-mediated pluripotency pathways, neuronal differentiation, and cell-cycle (Supplementary Fig. 4d, right panel).

DOT1L prevents premature differentiation of the PAX6-positive neural progenitor pool in the developing cortex in vivo[12]. The functional signature observed in NPC48h suggested that acute DOT1L inhibition might be sufficient to

bias the transcriptome from a stemness- to a differentiation-mediating transcriptional program. In line with this observation, we noticed a consistent decreased expression of a variety of neural stem cell markers in EPZ-treated NPC48h (Fig. 3c). To further substantiate this interpretation, we intersected our DEG set in NPC48h (EPZ vs DMSO treatment) with markers of neurogenic and neuronal cortical cell populations defined in two recent reports[31,32]. We observed that differentially expressed markers of neurogenic cell populations, mostly decreased in expression in our dataset, while differentially expressed markers expressed by fully differentiated neurons transcriptionally increased (Fig. 3d).

Together, these results indicate that DOT1L inhibition for 48 h was sufficient to deplete H3K79me2 on enriched loci genome-wide and to bring about mild yet functionally coherent transcriptional changes. Interpretation of the transcriptional response from a functional perspective suggests that DOT1L inhibition primes the transcriptome of NPC towards neuronal differentiation.

**Local epigenetic dynamics is induced by DOT1L inhibition**. The role of DOT1L as a chromatin writer demanded a thorough analysis of the association between transcriptional and chromatin alterations. In mESC and NPC48h, quantitative ChIP-seq revealed that DOT1L inactivation did not consistently affect the global levels of histone modifications other than H3K79me2 (Fig. 4a). Although EPZ treatment caused a decrease in H3K79me2 signal on every gene positively marked with this histone modification, the linear association of H3K79me2 depletion with transcriptional deregulation was weak in mESC (β = 0.027), and vanishingly small in NPC48h (β = 0.004) (Supplementary Fig. 5a). This observation indicated that DOT1L inhibition and subsequent reduction of H3K79me2 were not critical for the immediate expression of most genes.

We observed a difference in the mean expression level of transcriptionally deregulated genes. Specifically, upregulated genes tended to be lowly expressed, while downregulated genes tended to be highly expressed compared to control (Supplementary Fig. 5b). H3K27ac levels correlate with expression levels and recent studies suggest that H3K79me2 is important for maintaining H3K27ac enrichment on gene promoters and enhancers[24,25]. To verify whether H3K27ac signal was affected as a consequence of EPZ treatment, we performed differential analysis of H3K27ac peaks. We observed a few significant changes in the profile of H3K27ac peaks upon EPZ treatment compared to DMSO-treated samples, for both mESC and NPC48h. Log2 fold-change estimates of H3K27ac peaks overlapping the promoter of DEG showed a weak trend consistent with expression changes (Fig. 4b, upper panel). Notably, the effect size was stronger in NPC48h compared to mESC, despite a smaller number of genes being transcriptionally affected in the former cell type compared to the latter. A similar trend could also be observed for H3K4me3 (Fig. 4b, lower panel).

Annotation of H3K27ac peaks to overlapping/proximal genes revealed a weak genome-wide correlation between acetylation and transcriptional changes (Pearson correlation coefficient = 0.19 and 0.16 in mESC and NPC48h respectively) (Supplementary Fig. 5c). We observed a more evident loss of H3K27ac signal in a subset of genes with decreased expression upon DOT1L inhibition in NPC48h (Supplementary Fig. 5c, right panel).

Together, these data indicate that the genome-wide depletion of H3K79me2 did not result in a comparable global or local loss of H3K27ac. Our data showed that local epigenetic changes of active marks (e.g., H3K27ac, H3K4me3) were directly linked to transcriptional changes, as indicated by the small effect size and the specific association with deregulated genes.

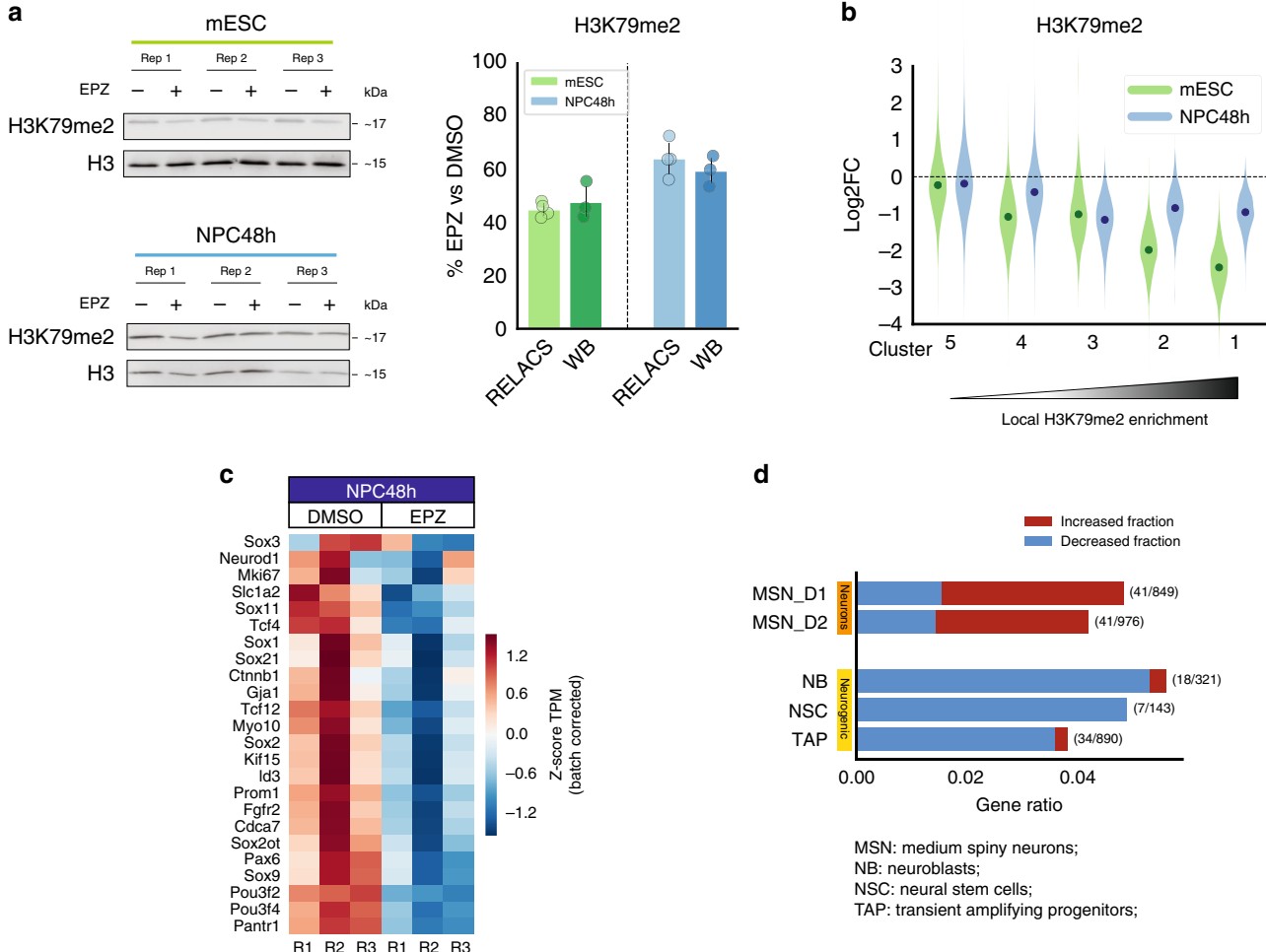

**Fig. 3 DOT1L inhibition for 48 h alters the transcriptome of NPC towards neuronal differentiation. a** EPZ5676 treatment for 48 h reduces the level of H3K79me2 in mESC and NPC48h. Left panel: immunoblotting of H3K79me2 and H3 (loading control) of EPZ-treated and DMSO-treated mESC (upper panel, green) and NPC48h (lower panel, blue). Right panel: barplot showing the global H3K79me2 signal for immunoblotting and RELACS ChIP-seq in EPZ-treated mESC and NPC48h, represented as a fraction of the respective H3K79me2 level in DMSO treated cells. For RELACS data, we calculated the ratios of uniquely and high-quality mapped reads (mapq > 5) and divided by the ratio for the respective inputs. $N = 3$ independent biological replicates were used for WB quantification, while for RELACS we show all pairwise comparisons across two biological replicates ($N = 4$ comparisons in total). Error bars represent $+/-1$ S.D. **b** Global fold-change of H3K79me2 in EPZ-treated cells compared to DMSO, over the five clusters identified in the differentiation analysis of Fig. 2d. High H3K79me2 local enrichment results in high loss upon EPZ treatment. **c** EPZ treatment decreases the expression of neural progenitor marker genes. Heatmap showing the z-score of batch corrected expression of various neuronal stem cell markers in DMSO and EPZ-treated samples. Triplicates for each treatment group are shown. Expression values are normalized using transcripts per million (TPM). Positive z-scores are in shades of red, while negative ones are in shades of blue. **d** Barplot showing the proportion of marker genes of three neurogenic cell types (NB, NSC, TAP) and two fully differentiated neuronal types (MSN_D1, MSN_D2)[31] that are differentially expressed in NPC48h following EPZ treatment (gene ratio, x-axis). Neurogenic marker genes are preferentially downregulated (blue), while marker genes of fully differentiated neurons are preferentially upregulated (red) upon EPZ treatment. Next to each bar, the gene ratio is explicitated as the number of marker genes that are differentially expressed in our dataset (numerator) over the total number of marker genes for each cell-type (denumerator).

**DOT1L inhibition associates with chromatin state signature.** To better characterize the transcriptional response to DOT1L inhibition in relation to the underlying chromatin states, we performed genome-wide chromatin segmentation. This method reduces the high dimensionality of the epigenomic dataset by assigning a unique chromatin state attribute to each genomic bin based on the combination of histone modification enrichment[33,34]. We used the chromatin segmentations of the control samples from mESC and NPC48h to measure the fraction of each chromatin state overlapping the promoter and the gene body of protein-coding genes genome-wide. We applied t-distributed stochastic neighbor embedding (tSNE) to visualize

the distribution of genes in a reduced space[35]. Mapping of the DEG set revealed a clear separation between upregulated and downregulated genes, which was consistent across cell types (Fig. 4c). Upon DOT1L inhibition, genes with a null, Polycomb repressed (H3K27me3) or bivalent (copresence of H3K4me3 and H3K27me3) promoter state were predominantly upregulated, while genes marked with an active promoter state (copresence of H3K4me3 and H3K27ac) tended to be downregulated (Fig. 4c).

To quantify the strength of association between chromatin states and transcriptional deregulation, we fitted a varying intercept model to estimate the expected transcriptional changes in each group of genes identified by the most represented

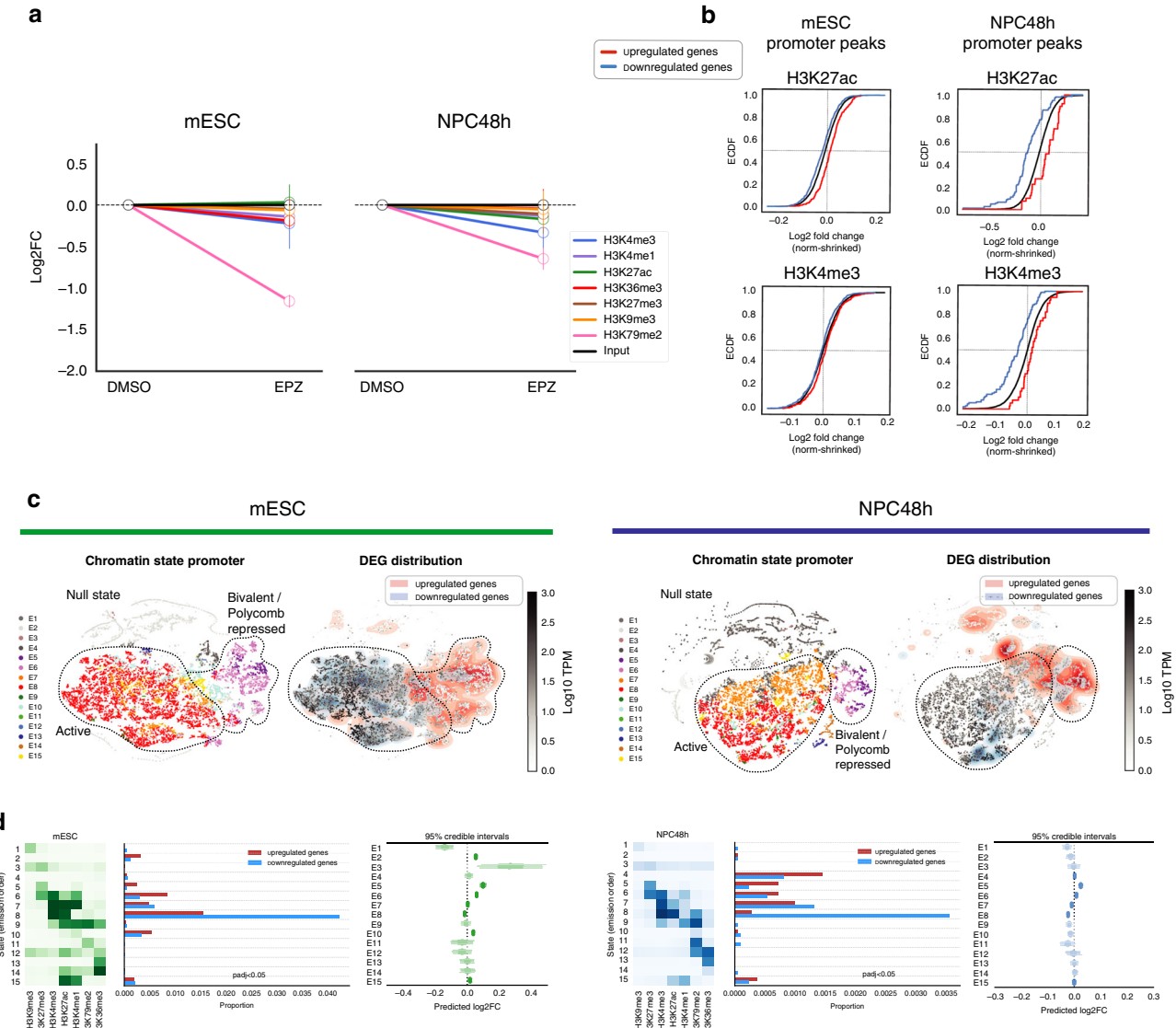

**Fig. 4 DOT1L inactivation results in local epigenetic changes and upregulation of Polycomb repressed genes in NPC48h. a** Log2 fold-changes of global scaling factors estimated via RELACS in EPZ-treated cells with respect to DMSO levels, for the seven histone modifications included in this study for mESC (left) and NPC48h (right). **b** Promoter-associated active marks change consistently with EPZ-induced transcriptional dynamics. Upper panel: empirical cumulative density function (ECDF) of log2 fold-change of H3K27ac (top) and H3K4me3 (bottom) peaks overlapping promoter (TSS−1000 bp, +500 bp) of differentially expressed genes (EPZ vs DMSO, adjusted *p*-value < 0.05) in mESC (left) and NPC48h (right). The red line shows the ECDF of log2 fold-change of peaks overlapping promoters of upregulated genes upon EPZ treatment, while the blue and black line depicts the same information for downregulated genes and all annotated genes respectively. Transcriptionally affected genes upon EPZ treatment show a corresponding gain/loss of H3K27ac and H3K4me3 on their promoters. The epigenetic response is evident in NPC48h, while is almost absent in mESC. **c** Dimensionality reduction (tSNE) of the chromatin state signature of protein-coding genes for mESC (left) and NPC48h (right). Genes are represented as dots. For each cell type, color-code is based on the most abundant chromatin state present in the promoter region on the left map. On the right map, color-code is a gradient mapping the expression level of each gene. A 2D kernel density plot was over-imposed to show the distribution of differentially expressed genes (adjusted *p*-value < 0.05) on the tSNE map. **d** Promoter chromatin state is weakly associated with transcriptional deregulation. For mESC (left) and NPC48h (right), heatmap summarizing the emission probability of the hidden Markov model, next to a histogram showing the proportion of differentially expressed genes (adjusted *p*-value < 0.05) classified according to the the most abundant promoter chromatin state. Next, a plot showing the expected log2 fold-change posterior distribution (95% credible interval) of each group of genes sharing the same most represented chromatin state in the promoter. Log2 fold-change distributions of states not associated with promoters are shaded. For each group, the expected mean expression log2 fold-changes is close to 0, suggesting that a small fraction of genes in each group is transcriptionally affected.

chromatin state present in the promoter region. Results showed that the estimated mean expression log2 fold-change in each gene group mildly deviated from 0, which suggested that the presence of any given chromatin state was not sufficient to induce transcriptional deregulation (Fig. 4d).

Together, this evidence suggests that DOT1L inhibition resulted in the derepression of silent genes and suppression of highly transcribed genes. Despite there being an association between transcriptional deregulation and chromatin states, only a small subset of genes was affected by DOT1L inhibition. This

prompted us to exclude that any specific combination of histone marks may be causally linked to the observed transcriptional phenotype. Instead, we hypothesized that the transcriptional changes may be mediated by mistargeting of transcription factors (TF) regulating the subset of transcriptionally affected genes.

**DOT1L block induces decreased enhancers accessibility in NPC.** To explore the hypothesis that DOT1L inactivation may affect specific DNA-binding TF, resulting in targeted gene expression changes, we profiled chromatin accessibility via ATAC-seq in mESC and NPC48h treated with DMSO or EPZ. Godfrey et al.[25] reported that accessibility of H3K79me2-marked enhancer and enhancer-associated H3K27ac decrease as a consequence of DOT1L inhibition in leukemia models. Therefore, we analysed how the accessibility profile was affected on enhancer regions genome-wide as a consequence of EPZ treatment during neural differentiation.

To study if the accessibility of enhancer regions was perturbed in mESC and NPC48h upon DOT1L inhibition, we performed differential analysis of open chromatin regions between treatment groups. Similar to previous assays, we observed minor alterations of the accessibility landscape, with very few regions reaching statistical significance. In mESC, DOT1L inhibition only results in minor changes in the accessible chromatin landscape, as indicated by the small fraction of total variance accounted for by the second principal component of ATAC-seq data (PC2 in Supplementary Fig. 6a) and the lack of correlation with gene expression (Supplementary Fig. 6b). In NPC48h, on the other hand, DOT1L inhibition corresponds to the highest source of variance (PC1 in Supplementary Fig. 6c). To determine open chromatin regions with high contribution to the first principal component (PC1), we selected 2000 peaks with the highest PC1 loadings, ranked on absolute value, and we visualized the fold-change distribution of enhancer regions (Fig. 5a). The results showed that intergenic and intronic enhancers tended to lose accessibility upon EPZ treatment. When we correlated dynamic accessible regions with expression changes of overlapping or proximal genes, we observed that loci with decreased accessibility were mostly associated with downregulated genes, regardless of enhancer status (Fig. 5a).

Together, these observations suggest that decreased chromatin accessibility on enhancer regions in response to DOT1L inactivation might contribute to the observed transcriptional downregulation in NPC48h.

**SOX2 enhancers are affected upon DOT1L block in NPC.** To identify candidate TF at chromatin regions with changed accessibility, we determined enriched sequence motifs on 2000 loci with the biggest loading in absolute values on PC2 and PC1, for mESC and NPC48h respectively. Regions with decreased accessibility (ATAC-Down) were enriched for motifs of pluripotency factors, such as NANOG and POU5F1 for mESC, and a range of POU/SOX core motifs for NPC48h (Fig. 5b).

The intersection of our dataset with previously published SOX2 and NANOG profiles indicated that the factors did not specifically bind ATAC-Down regions in mESC[36,37] (Supplementary Fig. 6d). The intersection of dynamic and unchanged accessible chromatin regions (background-ATAC) with publicly available SOX2 ChIP-seq data generated on brain-derived NPC[38] showed specific binding on open regions with decreased accessibility upon DOT1L inhibition (Fig. 5c). SOX2 ChIP followed by qPCR confirmed that DOT1L inhibition was associated with decreased SOX2 binding in a subset of target enhancers (Fig. 5d).

Taken together, our analysis shows that a subset of enhancers bound by SOX2 in vivo presents reduced accessibility and decreased SOX2 binding in NPC48h upon DOT1L inhibition. Since SOX TFs are critical regulators of neural progenitor pool maintenance and cell-fate specification[39–41], this may partly explain the biasing of NPC48h towards downstream differentiation.

Godfrey et al. reported on the association between affected enhancers upon EPZ treatment and loss of H3K79me2 and H3K27ac[25]. First, we investigated whether decreased accessibility was specifically associated with the presence of H3K79me2. We compared H3K79me2 density over dynamic chromatin regions and over unchanged regions that remain accessible in response to EPZ treatment. To this end, we also differentiated between intronic and intergenic regions (Supplementary Fig. 6e). Within the intronic class, we observed no significant difference between dynamic and unchanged enhancers (two-sided Mann–Whitney $U$ test, $p$-value $= 0.113$), but we observed a significant difference between intronic and intergenic enhancers (two-sided Mann–Whitney $U$ test, $p$-value $= 4.99 \times 10^{-7}$).

Next, we restricted our analysis to introns of protein-coding genes. We observed that 62% of protein-coding genes having at least one ATAC peak with decreased accessibility were marked with H3K79me2, while only 25 and 34% of protein-coding genes associated with ATAC-Up and background-ATAC regions were marked with H3K79me2 (Supplementary Fig. 6f). This indicated that reduced chromatin accessibility upon DOT1L inhibition mostly, but not exclusively, affected regions marked with H3K79me2 in intronic loci. However, the presence of H3K79me2 alone—and its consequent loss upon EPZ treatment—was not a discriminant factor for decreased accessibility.

To assess whether H3K27ac was affected on ATAC-Down enhancers depending on H3K79me2 presence[25], we visualized H3K27ac coverage on ATAC-Down enhancers high in H3K79me2, on ATAC-Up and ATAC-Down ATAC enhancers low in H3K79me2, as well as on background-ATAC regions. Results indicated that H3K27ac was not selectively decreased on enhancers in ATAC-Down regions as a consequence of H3K79me2 presence (Supplementary Fig. 6g).

In summary, these data indicated that, in our system, DOT1L inhibition affects binding of SOX2 to some target regions, but that loss of chromatin accessibility neither associate with depletion of H3K27ac nor is it strictly correlated to the occurrence of H3K79me2 enrichment.

## Discussion

Here we report on a comprehensive multi-omics study of in vitro neuronal differentiation and on the consequences of DOT1L inhibition for the differentiation process. Our study includes analysis of the quantitative dynamics of chromatin modifications during in vitro neuronal differentiation by the use of a quantitative and high-throughput ChIP-seq method. This is, to our knowledge, the first application of a quantitative strategy to a physiological differentiation setting, and it reveals that the epigenome of neuronal committed cells undergoes global histone modification changes with respect to the pluripotent precursor.

Various studies have documented a progressive chromatin condensation during mESC differentiation[4,42,43], but contrasting evidence has been collected regarding the extent and relevance of global histone modification changes for cellular differentiation. For example, Ugarte et al. describe a progressive decrease in nuclease sensitivity during hematopoietic differentiation but fail to detect any significant global changes in histone modifications levels through immunoblotting assessing H3K4me3, H3K27ac, H3K16ac, H4K20me1, H3K36me3, H3K27me3, H3K9me2, and

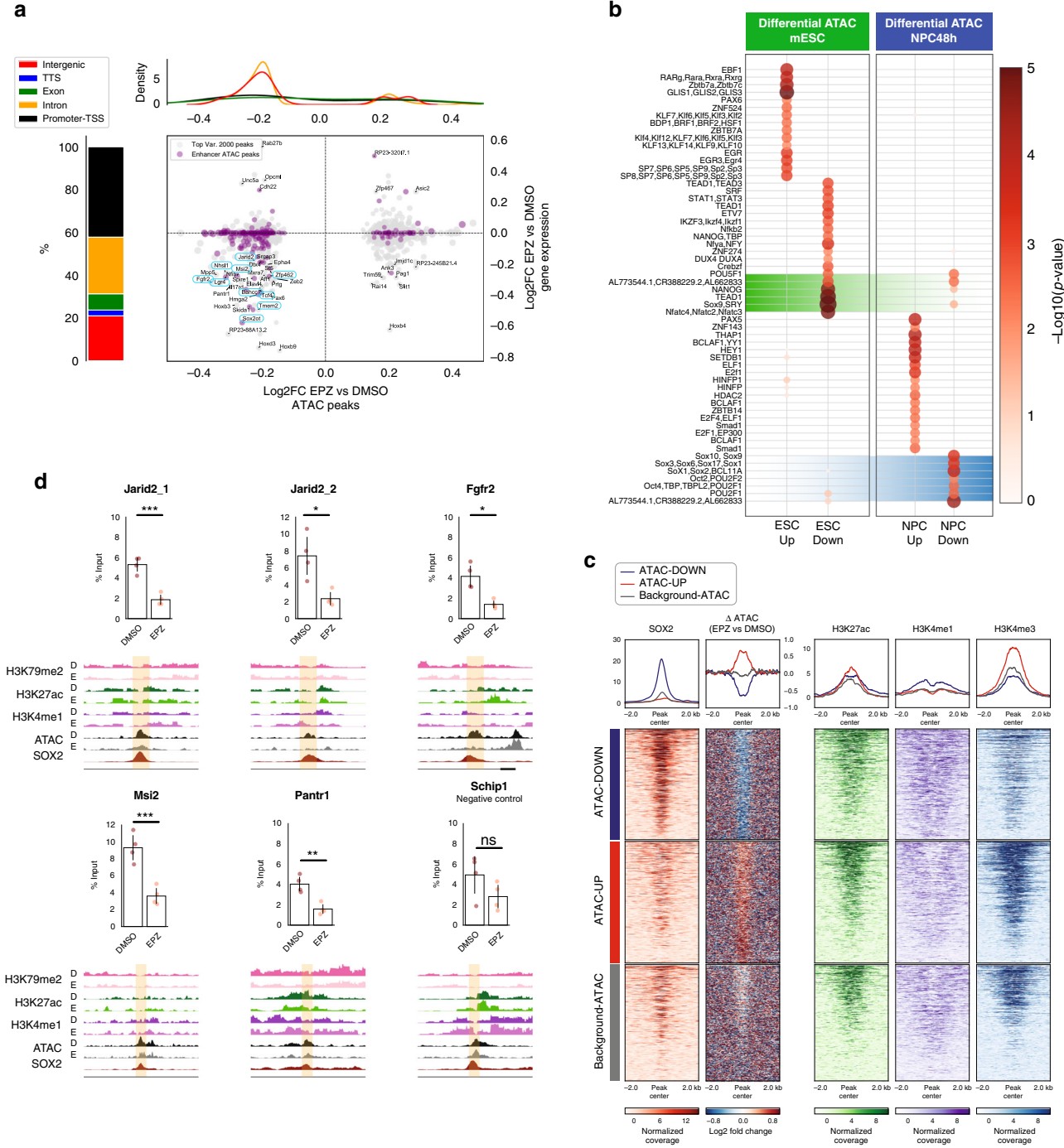

H3K9me3[42]. Efroni et al. characterize global transcriptional and epigenetic changes during mESC-derived NPC differentiation. Their evidence, based on immunoblotting and imaging, suggests that both global RNA levels and active histone modification abundances (e.g., H3K4me3) are decreased in differentiated cells compared to the embryonic precursors[4].

In contrast to these previous studies, here we use a quantitative ChIP-seq protocol (RELACS) to estimate global histone modification changes. Our main finding is that only H3K27ac and H3K79me2 levels change globally, in opposite directions, during in vitro neuronal differentiation, through genome-wide acting mechanisms.

Biologically, these results show that the progressive chromatin condensation observed during in vitro neuronal differentiation

mostly follows from a genome-wide deacetylation process, while a smaller contribution may come from the global accumulation of repressive histone modifications (e.g., H3K9me3)[44]. Loss of H3K27ac is consistent with chromatin condensation[1], as H3K27ac is a mark associated with loose chromatin packaging and is known to be highly abundant in mESC[44,45].

Secondly, our data show that H3K79me2 increases globally during neuronal differentiation in vitro. Michowski et al.[46] have recently shown that DOT1L localization and activity is controlled by CDK1 in mESC, causing global epigenetic effects during differentiation. The proposed mechanism is consistent with the global H3K79me2 increase that we observe during neuronal differentiation. Developmental gain of local enrichment of H3K79me2 associates with transcriptional activation of genes

**Fig. 5 DOT1L inhibition decreases the accessibility of SOX2 target enhancers and SOX2 binding in NPC48h. a** Characterization of dynamic ATAC peaks. Left panel: stacked barplot summarizing the genomic distribution of the 2000 most dynamic ATAC peaks (blue: TTS, green: exon, yellow: intron, black: promoter-TSS). Central panel: scatterplot showing the association between the log2 fold-change of dynamic ATAC peaks (*x*-axis) and the log2 fold-change of genes overlapping or proximal to at least one dynamic ATAC peak (*y*-axis) upon EPZ treatment in NPC48h. ATAC peaks overlapping enhancer regions are shown in purple. Gene symbols are shown for differentially expressed genes upon DOT1L inhibition. Genes associated with dynamic ATAC peaks that are downregulated in both mESC and NPC48h are highlighted in boxes. Top panel: density plot of the accessibility log2 fold-change of ATAC peaks overlapping enhancer upon EPZ treatment, stratified according to genomic annotation. **b** Differential motif analysis on differentially accessible ATAC peaks upon EPZ treatment in mESC (left) and NPC48h (right). The size and color of each dot is proportional to −log10(*p*-value) associated with the enriched motif. **c** SOX2 preferentially binds in vivo to regions with reduced accessibility upon EPZ treatment. Metaprofile and heatmap of SOX2 binding profile in brain-derived NPC[38] over open regions losing accessibility (ATAC-Down), gaining accessibility (ATAC-Up) and over unaffected regions (background-ATAC) upon EPZ treatment in NPC48h. Metaprofile and heatmap of the corresponding log2 ratio of EPZ vs DMSO ATAC-seq signal, H3K27ac, H3K4me1 and H3K4me3 input subtracted coverage of DMSO treated sample. **d** SOX2 binding decreases on enhancers with decreased accessibility upon DOT1L inhibition compared to control. For each gene, % of immunoprecipitated input is shown (*y*-axis) for SOX2 ChIP on DMSO and EPZ-treated NPC48h. Enhancers overlapping with or proximal to *Jarid1*, *Fgfr2*, *Msi2*, and *Pantr1* were tested. One enhancer overlapping *Schip1* showing no dynamic accessibility upon DOT1L inhibition was chosen as negative control. Four independent biological replicates were compared for each locus (shown as dots). Error bars represent standard deviation. Significance for differential SOX2 binding was computed with a one-sided paired student *t*-tests. *$p$-value < 0.5; **$p$-value < 0.05; ***$p$-value < 0.005; n.s: not significant. Exact $p$-values: Jarid2_1 = 0.0010, Jarid2_2 = 0.0283, Fgfr2 = 0.0134, Msi2 = 0.0009, Pantr1 = 0.0025, Schip1 = 0.2664.

critical for neuronal differentiation, although the accumulation may be a byproduct of transcription. Previous reports have shown that the ectopic localization of DOT1L to the promoter region of few target genes, and the consequent accumulation of H3K79me2, is not sufficient to initiate transcription[47].

As a consequence of H3K79me2 global increase, we investigate the relevance of DOT1L methyltransferase activity for the establishment of chromatin and transcriptional states in mESC and their differentiated progeny, NPC48h. The third main result of this study shows that DOT1L inactivation affects gene expression in a targeted manner, despite the genome-wide depletion of H3K79me2. Our results also indicate that the presence of H3K79me2 is neither critical for the deposition of other histone modifications, nor necessary for sustaining the expression levels of most genes. We observe depletion of H3K27ac upon EPZ treatment, but this does not follow the global decrease in H3K79me2. Locally, however, loss of H3K27ac on enhancers and promoters alike correlates with transcriptional downregulation, and it is mirrored by a corresponding decrease in H3K4me3 on promoters. These data argue against the hypothesis that H3K79me2 is generally critical to preserve H3K27ac from deacetylase complexes in our model[24]. In NPC48h, DOT1L inhibition seems to prime the transcriptome of neural progenitors towards neuronal differentiation. Although the transcriptional deregulation is moderate in effect size, epigenetic changes are associated with expression dynamics, suggesting that the altered transcriptional state induced by DOT1L inactivation might be epigenetically memorized and preserved through mitotic divisions.

Interestingly, transcriptionally deregulated genes present a coherent chromatin signature. Our data indicate that DOT1L inactivation associates with upregulation of genes with a repressed, poised or null promoter state, and downregulation of highly expressed genes marked with active histone modifications. Based on the data presented in this work, it is tempting to hypothesize that the transcriptional upregulation upon DOT1L inhibition observed in the mammalian system may result from impaired targeting of repressive complexes. The cause of this may either reside in the altered H3K79me distribution, as in the yeast model, or it may indirectly follow from the selective downregulation of highly transcribed genes coding for repressive proteins (e.g., *Jarid2*, *Zfp462*)[48–51].

Finally, our study indicates that the targeted transcriptional response to DOT1L inactivation may be partly explained by decreased accessibility of active enhancers bound by critical TF. Enhancers are state-specific and change rapidly during differentiation[52]. As EPZ treatment seems to bias NPC towards downstream differentiation, this may account for the greater effect size that we observe in chromatin accessibility dynamics and local H3K27ac changes in NPC48h compared to mESC upon DOT1L inhibition. Whereas in NPC48h, DOT1L inhibition results in decreased accessibility at chromatin regions bound by SOX2 in vivo, the reduced chromatin accessibility is not accompanied by depletion of H3K27ac. In this light, our data partly contrast with the model advocated by Godfrey et al.[25], which establishes a causal link between the presence of H3K79me domains, preservation of H3K79me-rich enhancer activity and H3K27ac levels. Godfrey et al.[25] recently reported a class of enhancers dependent on H3K79 methylation, where the frequency of enhancer-promoter interaction is disrupted upon DOT1L inhibition. Consistently, we observe that EPZ treatment induces a decrease in accessibility of a subset of intronic and intergenic enhancers. Our data also indicate that H3K79me2 enrichment does not discriminate between dynamic and non-dynamic open regions in our system, suggesting that DOT1L inhibition may affect a subclass of enhancers independently from H3K79 methylation. While local H3K79me2 enrichment does not specifically associate with decreased accessibility of SOX2-bound enhancers, the relation between H3K79me2 global increase during differentiation and the effects of DOT1L inhibition on SOX2-enhancers remains obscure. Further studies should be conducted to elucidate whether the global accumulation of H3K79me2 bears important consequences for the enhancer activity of neural committed cells.

In conclusion, our findings agree with the model proposed by Godfrey et al. in that DOT1L inhibition results in decreased accessibility of some H3K79me2-positive enhancers. In our system, though, we observe a specific response that pertains to a subset of regulatory regions bound by sequence-specific transcription factors (e.g., SOX/POU). The closure of these enhancers may be responsible for the transcriptional decrease of highly expressed genes conferring stemness to progenitors. In addition, here we present evidence that could explain transcriptional increase upon DOT1L inhibition. We hypothesize that decreased expression of cell-type-specific transcripts coding for proteins with repressive functions (e.g., *Jarid2*, *Zfp462*) may result in the derepression of silent genes localized on facultative heterochromatic regions.

## Methods

**mESC culture and in vitro neuronal differentiation**. mESC *Dot1l-HA-FLAG* (constructed and purchased from inGenious Targeting Laboratory) were cultured on inactivated MEF for 3 passages (p3) and from p4 onward on gelatin-coated plates (medium: 82% DMEM (Thermo Fisher, US), 15% FBS (Thermo Fisher), 1% Glutamax (Thermo Fisher), 1% PSN (Thermo Fisher), 1% NEAA (Thermo Fisher) + LIF (Sigma) (dilution = 1/1000) + β-Mercapto-EtOH (Thermo Fisher) (dilution = 1/500)). Feeder-free mESC were treated with either EPZ5676 (Hycultec) (10 μM), or DMSO (Thermo Fisher) (dilution = 1/1000) for 48 h.

mESC were differentiated in vitro towards NPC48h according to Bibel et al.[26]. In brief, feeder-free mESC were trypsinized and dissociated to create a single-cell suspension. Cells were used to form cell aggregates (CA) on non-adherent (Grunier) plate ($4 \times 10^6$ single cells per plate; medium: 87% DMEM, 10% FBS, 1% Glutamax, 1% PSN, 1% NEAA + β-Mercapto-EtOH (1/500)). Four days after CA formation, CA were exposed to RA (7.5 μM) for 4 days. CA were dissociated into single cells and seeded on PORN/LAMININ coated 6-well plates and grown in N2 medium for neuronal differentiation. At this stage, cells were treated either with EPZ5676 (10 μM) or DMSO (1/1000) for 48 h. At treatment completion, NPC48h were collected for downstream processing.

**RELACS ChIP-seq**. RELACS protocol was carried out according to Arrigoni et al.[5]. Cells were fixed in 1% formaldehyde for 15 min. The reaction was quenched with 125 mM glycine for 5 min, followed by two washings with DPBS + proteinase inhibitor cocktail. Cell nuclei were isolated following Nexson protocol[53] and permeabilized with 0.5% SDS. Chromatin was digested in situ using restriction enzyme CviKI-1 (NEB, R0710L) and barcoded using RELACS custom barcodes (4 bp UMI + 8 bp RELACS barcode, see Supplementary Table 1 for details). Nuclei from each sample were burst via sonication to extract the barcoded chromatin fragments and pooled into a unique tube. A single immunoprecipitation (IP) reaction for all samples included in this study was carried out on IP-star according to[5] (see Supplementary Table 2 for antibodies details). Immunoprecipitated chromatin was used for Illumina library preparation (NEBNext Ultra II DNA Library Prep Kit) and sequenced on HiSeq 3000 Illumina machine (paired-end, read length 75 bp).

**RNA-seq**. RNA was extracted using RNAeasy Mini Kit (Qiagen). Libraries were generated using the NEBNext Ultra RNA Library Prep Kit, following manual's instructions. Libraries were sequenced on a HiSeq 3000 Illumina machine (paired-end, read length 150 bp).

**ATAC-seq**. ATAC-seq libraries were generated according to[54]. In brief, ~50,000 cells were washed in ice-cold PBS and incubated in the transposition reaction mix (Nextera DNA Sample Preparation Kit). Transposed DNA was purified (MiniElute Kit, Qiagen) and PCR amplified for 5 cycles. We determined the number of additional PCR cycles via qPCR according to[54]. Libraries were sequenced on a HiSeq 3000 Illumina machine (paired-end, read length 75 bp).

**Immunoblotting**. mESC or NPC48h were lysed in RIPA buffer (1% NP-40, 1% SDS, 0.5% sodium deoxycholate diluted in Phosphate Buffered Saline, PBS). Cells were centrifuged (10 min, 13,000 rpm) and the supernatant collected. Protein concentrations were determined with Bradford reagent (Bio-Rad). Fifteen micrograms of protein extract were loaded with 5x Laemmli buffer on Mini Protean TGX gels (Bio-Rad) and run at 100 V for 1.5 h. Proteins were transferred to PVDF membranes (Trans-blot Turbo Transfer Pack) using the Trans-blot Turbo Transfer System (both from Bio-Rad) following manufacturer's instructions. Membranes were blocked with 5% BSA in TBS-T (blocking buffer) for 1 h and incubated overnight with primary antibodies (diluted in blocking buffer). Membranes were washed, incubated with secondary antibodies for 1 h and detected using ECL or Femto substrates (Thermo Scientific) and LAS ImageQuant System (GE Healthcare, Little Chalfont, UK). Antibodies details are listed in Supplementary Table 3. For densitometric analyses, ImageJ software was used[55]. The code used for the analysis of densitometric data is available at https://github.com/FrancescoFerrari88/code_DOT1L_paper/blob/master/figure_3/NOTEBOOKS/3a-3b_EPZ_vs_DMSO_H3K79me2_GLOBAL_BULCK_%26_LOCUS-SPECIFIC_CHANGES.ipynb.

**ChIP-qPCR**. ChIP-qPCR was performed as described previously[23] using following deviating buffer recipes: lysis buffer (50 mM Tris at pH 8.0, 10 mM EDTA, 0.1% (w v$^{-1}$) sodium dodecyl sulfate (SDS), and 1× Protease inhibitor) dilution buffer (20 mM Tris at pH 8.0, 150 mM NaCl, 2 mM EDTA, 1% (v v$^{-1}$) Triton X-100, 0.1% (w v$^{-1}$) SDS, and 1× Protease inhibitor). NPC48 cells grown on 1 × 10 cm dish (~6 Mio cells) treated for 48 h with 10 μM EPZ5676 or DMSO were used for one SOX2- and control-ChIP. Two micrograms of the following antibodies were used per ChIP: anti-SOX2 (Santa Cruz, sc-365964), mouse IgG (Diagenode, C15400001-15). Significance of SOX2 binding to the specified loci between EPZ5676 treatment and control was calculated with a one-sided paired student *t*-tests. For that, four independent biological replicates were compared. The following primers were used for detection:

*Jarid2_1* (for TGTGCAGGCTGTAGCCATTA, rev GGGGACCCACAAGCAC TACAA), *Jarid2_2* (for CTGGAAGCCTGTGTGGAGAG, rev GGGGAGAATG GATGGAGCTG),

*Fgfr2* (for CCACTTCTTGGGGCGCTAAA, rev CCTCTCGCTCTCGTTGGA AT),

*Msi2* (for ACCGCACACACAAATCATGC, rev GGCAGGAAAACGGTGTA GGA),

*Pantr1* (for CTAATTCCCGGGCTCAAACG, rev GGTGGGGAACTCGCTA CAG),

*Schip1* (for TCAGCGGAACAGCATCCATT, rev CTAGGACGCAGACCCC AAAG).

**Bioinformatics analysis**. All sequencing data were processed with snakePipes (v. 1.1.1)[56]. Relevant parameters used for each experiment and summary QC are available at https://github.com/FrancescoFerrari88/code_DOT1L_paper/tree/master/multiQC_ConfigParameters. Mapping was performed on mouse genome build mm10 (GRCm38). For ChIP-Seq and ATAC-seq, high quality and uniquely mapping reads were retained (mapq > 5). RELACS custom barcodes were designed with integrated UMI, so duplicate removal was performed using UMITools[57], while a standard deduplication was applied for ATAC-seq reads. We use gencode M18 as a reference gene model throughout all analysis. For ChIP-seq and ATAC-seq data, snakePipes also provided candidate peak regions using MACS2 (v. 2.2.6) using default parameters.

Differential analysis for RNA-seq was carried out using DESeq2 (v. 1.22.1)[58] on count matrices output from snakePipes (featureCounts, v. 1.6.4). We used a linear model controlling for batch effects (e.g., ~batch + treatment) and we applied apeglm log2 fold-change shrinkage[59].

We estimate fold-changes for each histone modification on annotated genomic features known to associate with local histone PTM enrichment (H3K4me3, H3K27ac, H3K4me1: narrow promoter (TSS ± 1 kb); H3K79me2, H3K27me3, H3K9me3: extended promoter (TSS −1kb, +3 kb); H3K36me3: transcription termination site (TTS −3 kb, +0.5 kb)).

Global differential ChIP-seq analysis was carried out after applying RELACS specific empirical normalization by computing empirical log-fold changes across conditions (see "RELACS estimation of global histone modification changes"). Traditional differential ChIP-seq and ATAC analysis was performed on consensus peak sets, coverage was computed using deepTools' multiBamSummary[60] and differential regions identified via DESeq2. We eventually applied normal log2 fold-change shrinkage. Peaks were annotated with Homer (v. 4.10)[61] and UROPA (v. 3.1.0)[62]. We use GimmeMotifs (v. 0.13.1) for motif enrichment and differential motif analysis[63]. Metaprofile of ChIP-seq and ATAC-seq signals were generated with deeptools (v. 3.1.2)[60] and deepStats (v. 0.3.1)[64]. The heatmap combining ChIP-seq and RNA-seq data was generated with Ultraheatmap (v. 1.3.0)[65].

Multiple factor analysis was done using FactoMineR (v. 1.41)[66]. The algorithm was run on a matrix of shape 4 (samples) × 3500 (features). As features, we included the top 500 most variable 2 kb loci for each of the seven histone modifications (feature groups), selected after applying variance stabilizing transformation to the counts matrix. We used scikit-learn (Python module) (v. 0.19.1) for PCA and tSNE, while linear modeling was performed using sklearn and statsmodels (v. 0.9.0). GO enrichment analysis and pathway analysis were performed using clusterProfiler (v. 3.10.1)[67].

We used ChromHMM (v. 1.15)[33] with default parameters for chromatin segmentation. We trained two independent models for each cell type on the DMSO treated samples. We then used these models to perform the segmentation in the respective cell types for both treatments (EPZ and DMSO).

We compute the chromatin state signature of protein-coding genes in mESC and NPC48h according to[35]. For each gene, we identify potentially used transcripts by intersecting annotated TSS with H3K4me3 peaks. If a gene does not overlap with any H3K4me3 peak, we consider the full gene annotation. For each candidate gene, we then compute the fraction of overlap between each chromatin state segment in the control sample with the promoter region (TSS−1kb, +500 bp) and with the full gene body. In this way, each gene is identified by a vector of length 30 (15 states for the promoter + 15 states for the gene body). A matrix of shape g (number of genes per cell) × 30 is eventually used for dimensionality reduction by applying tSNE[68].

Bayesian linear modeling was performed using pymc3 (v. 3.6)[69]. The expected log2 fold-change for each group of genes (i) identified by the most represented chromatin state present in the promoter regions was identified by fitting the following hierarchical linear model:

$$Log2FC \sim N(\mu, \sigma)$$
$$\mu \sim \alpha[i]$$
$$\sigma \sim Exp(\lambda = 1)$$
$$\alpha[i] \sim N(\mu', \sigma') \qquad (1)$$
$$\mu' \sim N(0, 1)$$
$$\sigma' \sim Exp(\lambda = 1)$$

All visualizations were generated in Python (v. 3.6) and R (v. 3.5).

**RELACS estimation of global histone modification changes**. To estimate global histone modification changes, first we demultiplexed fastq files on RELACS custom barcodes. Then, for each sample, we divided the number of uniquely and high-quality mapped read-pairs (mapq > 5) coming from a ChIP of interest by the total number of read-pairs coming from the respective input. For estimating global histone modification changes, we considered the total number of mapped reads genome-wide. Pairwise quantitative comparisons between samples were computed as log2 ratio between input-normalized total mapped read counts.

Local changes were estimated in the same way, by repeating this procedure for each individual bin of interest.

**Statistics and reproducibility**. All sequencing experiments presented in this paper have been conducted once, with the appropriate biological replicates included in the experimental run. Immunoblotting was replicated at least twice, with each replicate including all biological replicates shown in the study. SOX2 ChIP followed by qPCR was performed once.

**Reporting summary**. Further information on research design is available in the Nature Research Reporting Summary linked to this article.

## Data availability

SOX2 binding profiles in brain-derived NPC were retrieved from the following public GEO repository: GSE90559. SOX2 and NANOG binding in mESC were retrieved from public GEO entries GSM1050291 and GSM1847493 respectively. Raw data and normalized bigWig tracks generated in this work were deposited to GEO and are available for download using the following accession number: GSE135318. All other relevant data supporting the key findings of this study are available within the article and its Supplementary Information files or from the corresponding authors upon reasonable request. Original immunoblotting images are provided as supplementary Source Data file. A reporting summary for this Article is available as a Supplementary Information file. Source data are provided with this paper.

## Code availability

The fully reproducible and documented analysis is available on github at github.com/FrancescoFerrari88/code_DOT1L_paper, as Jupyter notebooks and R scripts.

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

## Acknowledgements

We would like to thank Ulrike Bönisch, Chiara Bella, Katrin Großer, and Steffen Wolter for providing essential support for the generation of the sequencing data. We thank Yaarub Musa and Gerhard Mittler for the fruitful discussions on the research project. Special thanks to Devon Ryan and Leily Rabbani for providing the scripts needed for RELACS demultiplexing and for integrating the RELACS workflow into the publically available NGS processing pipeline (snakePipes). We thank Alejandro Villarreal for help with mESC and NPC48h cultures and ATAC-seq. The useful comments and suggestions by Dr. Steven Johnson and one anonymous reviewer are gratefully acknowledged. This research was funded by the Deutsche Forschungsgemeinschaft (DFG, German Research Foundation): 322977937/GRK2344 and SFB 992 (Medical Epigenetics).

## Author contributions

T.M. and T.V. conceived the study. L.B. and F.F. performed in vitro differentiation of mESC to NPC48h. L.A. and F.F. executed the RELACS protocol and generated the ChIP-seq data. L.B. and L.A. generated the RNA-seq data. E.T. generated the ATAC-seq data. H.F. performed the SOX2-qPCR, while A.I. and L.B. generated the immunoblotting results. F.F. analyzed the data and wrote the manuscript, with inputs from T.M and T.V. T.M. supervised the analysis of F.F.

## Funding

## Competing interests

The authors declare no competing interests.

## Additional information

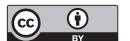

