## [Peer Review File · Nature Communications]

Reviewers' comments:

Reviewer #1 (Remarks to the Author):

In this study, using a direct comparative method, the authors profiled changes in histone modifications at the stage of mESCs and in-vitro neuronal differentiation, NPC48h. The authors also assessed how histone marks and gene expression (in both mESCs and NPCs), as well as chromatin accessibility (in NPCs only), are effected upon the DOT1L inhibition.

There are three main findings:

1. H3K27ac decreased, and H3K79me2 increased in NPCs compared to mESCs
2. DOT1L inhibition induced NPCs to a more mature state
3. DOT1L inhibition reduced the accessibility of intronic enhancers that contain SOX2 binding motifs

Overall, the manuscript is describing the results from the through bioinformatic analyses. However, there are a lot of analyses and plots that showcase the same result in different ways, making the manuscript more complicated than it has to be. The authors should think about the main message that is to be the focus of this manuscript. They should move some of the figure panels into the supplementary and rewrite results concisely with logical connections. The main messages should be delivered clearly with proper conclusions. Meantime, parts of the discussion contain appropriate conclusions that the authors can take them out and move to the result. Reorganization of the figures and rewriting of the manuscript will significantly strengthen the overall presentation of their analyses/findings.

The finding that SOX2 binding sites are affected by DOT1L inhibition is interesting; however, this message is buried underneath layers of information that did not develop fully. Along this line, the title of the paper is not correctly describing the overall findings of this manuscript but emphasizing only a small fraction of the final figure panel. I understood that it contains a novel finding that could connect DOT1L function to SOX2-enhancer regions, and the authors did comprehensive bioinformatics to reach to it. However, if it is the central message of the paper, the authors should perform a mechanistic study in addition to validation experiments. For example, how DOT1L affect SOX2 activity only in a subset of SOX2 targets? Is it NPC48h specific? Or also happens in mESCs? As the authors observed more DEGs in mESCs upon DOT1L inhibition, it will be essential to test whether they could see more broad changes in chromatin accessibility also in mESCs after DOT1L inhibition and connect them to SOX2 activity as well as transcription changes. Does activation of this SOX2-enhancer function (e.g., CRISPR activation) in NPC48h reverse the effect of DOT1L inhibition?

Here are the suggestions for each main results:

First part: epigenetic and transcriptional changes during ESCs neuronal differentiation

Global profiling of seven histone modifications related to transcription states in ESCs and NPC48h showed H3K27ac loss and H3K79me2 gain. These two modification changes are not linked to each other. This conclusion should be clearly described in the result compared with the previous finding (Godfrey et al.). This could strengthen the result from the RELACS based ChIP-seq assay.

In Figure 2d/e, the authors focused on H3K79me2 and how this mark correlates with gene expression. Only genes in cluster 3 showed an increase in H3K79me2 downstream of TSS. These genes were also upregulated in NPCs compared to mESCs and have a neuronal development signature. The authors can elaborate more on this result. Is the accumulation of H3K79me2 causative for the upregulation of these genes or just a byproduct of their increased expression? Is the expression of the genes in cluster 3 affected upon DOT1L inhibition? The authors mentioned this a bit in Figure 3b (cluster 3), but the message was not well delivered in the result, and the most relevant conclusion is in the discussion.

H3K27ac and H3K79me2 show opposite trends during neuronal differentiation, with H3K27ac decreasing in NPCs compared to mESCs. In order "to explore candidate mechanisms accounting for these histone modification changes, the authors investigate the transcriptional dynamics of genes coding for epigenetic enzymes". They conclude that since 4 out of 13 expressed HDACs show increased expression in NPCs, this could explain the decrease in H3K27ac levels. It would be interesting to use a proteomic study conducted in similar systems and test their findings. The authors also described 2 HATs changes in NPCs. It seems Kat6a is mainly involved in K9ac, and Hat1 is reported to be involved in the acetylation of newly synthesized cytoplasmic histones. What is the exact connection between 2 HATs (Kat6a and Hat1) and H3K27ac decrease?

Second part: epigenetic and transcriptional changes upon DOT1L inhibition

Most of all, the authors did not specify why they used a 10nM concentration of the inhibitor. Is it cited somewhere? Did the authors test different concentrations?

Figure 3a western blot seems to have differential loading of the H3 control protein for the NPC48h conditions. Is this accounted for the WB quantification? If DOT1L is the main focus of this study, which, as stated in the intro, regulates H3K79me1,2,3 - then immunoblots of H3K79me1 and H3K79me3 after EPZ treatment would be useful to show.

Considering the modest magnitude and numbers of the gene expression changes upon EPZ treatment in NPCs, is this enough to conclude that DOT1L inhibition mediates a shift in the transcriptional program of the progenitors? Are there any publicly available data showing a similar trend with DOT1L KD in NPCs?

Third part: chromatin accessibility changes upon DOT1L inhibition and Sox2

PCA plot and hierarchical clustering for gene expression analysis upon EPZ treatment show separation only between replicates and not between treatments (SFig.3a). PCA plot for ATAC-seq data in NPCs shows an excellent separation between drug and vehicle treatment. The authors should clearly describe their results and add interpretation. Is it a technical issue with the experiment, or is there a biological explanation for that?

While the H3K79me2 level is correlated with differentiation, inhibition of its writing enzyme facilitates neuronal differentiation by reducing SOX2 accessibility to intronic, and intergenic enhancers. What is the function of the global H3K79me2 increase during differentiation? What is the relationship between H3K79me2 enrichment and SOX2 binding? Is it directly related to the H3K79 methylation level and/or a byproduct of the non-enzymatic effect of the DOT1L?

Reviewer #2 (Remarks to the Author):

In this tour-de-force manuscript, Ferrari et al perform a multi-omic analysis of epigenetic, transcriptional and chromatin changes that occur during neuronal differentiation from mESCs to NPCs. They demonstrate that two of the seven histone marks they analyze H3K27ac and H3K79me2 surprisingly change in opposite directions (with minimal change in the other histone marks), differentiate between global and local epigenetic changes, and examine the effect that inhibition of DOT1L in mESC and mESC-derived NPC lines has on these cells, showing that inhibition of DOT1L is sufficient to initiate neuronal differentiation.

This study and analysis are excellently executed, the findings are novel and contribute greatly to the field and I recommend it for publication in Nature Communications with the following non-experimental, minor modifications/revisions:

- 1) On line 23,, define "MLL" as it is the first time you are using the term.
- 2) The second half of the compound sentence ending on line 70 is not a sentence and needs a

verb.

3) Line 73 change "to detect" to "detection of."

4) Line 95, get rid of "e.g." in the sentence.

5) The bulk of the RESULTS section needs to be changed to the past tense instead of the present tense, when explaining what the researchers did. Interpretation of the data can remain in the present tense, but the experiments and analysis need to be in the past tense.

6) Line 136, the outermost parenthesis need to be changed to brackets "[]" while the inner ones can remain curved.

7) Line 145, tell us which TSS and TTS size ranges were used for each modification (like you do on line 199) and why those regions were chosen, e.g. "since such and such histone mark is associated with proximal promoter regions...", or "since such and such is associated with the ends of coding regions...".

8) Line 193, the word "Eventually" is the wrong word and could be omitted completely or replaced with a word like "furthermore" depending on what exactly you wish to convey.

9) Line 258, same as above for "Eventually."

10) Line 279, remove the "to" in the phrase "signal equals to 47.8%..."

11) Line 408, change "to dissect" to "the dissection of."

12) Line 458, "enhancer" changed to "enhancers."

13) Line 482, change it so it reads, "affected on ATAC-Down enhancers depending on H3K79me2 level (26), we visualized..."

14) Line 502, there is an extra "s" in the sentence.

15) Line 592, change it so it reads "In addition, here we present the first evidence..."

Again, the experiments and analysis in this manuscript are impressive and well written, and I recommend publication in Nature Communications.

Steven M. Johnson. Ph.D.

Detailed response to reviewers

Reviewer #1 (Remarks to the Author):

In this study, using a direct comparative method, the authors profiled changes in histone modifications at the stage of mESCs and in-vitro neuronal differentiation, NPC48h. The authors also assessed how histone marks and gene expression (in both mESCs and NPCs), as well as chromatin accessibility (in NPCs only), are effected upon the DOT1L inhibition.

There are three main findings:

1. H3K27ac decreased, and H3K79me2 increased in NPCs compared to mESCs
2. DOT1L inhibition induced NPCs to a more mature state
3. DOT1L inhibition reduced the accessibility of intronic enhancers that contain SOX2 binding motifs

Overall, the manuscript is describing the results from the through bioinformatic analyses. However, there are a lot of analyses and plots that showcase the same result in different ways, making the manuscript more complicated than it has to be. The authors should think about the main message that is to be the focus of this manuscript. They should move some of the figure panels into the supplementary and rewrite results concisely with logical connections. The main messages should be delivered clearly with proper conclusions. Meantime, parts of the discussion contain appropriate conclusions that the authors can take them out and move to the result.

Reorganization of the figures and rewriting of the manuscript will significantly strengthen the overall presentation of their analyses/findings.

We would like to thank the reviewer for suggesting a streamlined presentation. The main findings from our study are correctly identified, but we would like to add the observation that DOT1L inhibition induces a chromatin-state specific response in gene expression (Figure 4). We have thoroughly revised both text and figures to communicate these points more concisely.

The finding that SOX2 binding sites are affected by DOT1L inhibition is interesting; however, this message is buried underneath layers of information that did not develop fully. Along this line, the title of the paper is not correctly describing the overall findings of this manuscript but emphasizing only a small fraction of the final figure panel. I understood that it contains a novel finding that could connect DOT1L function to SOX2-enhancer regions, and the authors did comprehensive bioinformatics to reach to it. However, if it is the central message of the paper, the authors should perform a mechanistic study in addition to validation experiments. For example, how DOT1L affect SOX2 activity only in a subset of SOX2 targets? Is it NPC48h specific? Or also happens in mESCs? As the authors observed more DEGs in mESCs upon DOT1L inhibition, it will be essential to test whether they could see more broad changes in chromatin accessibility also in mESCs after DOT1L inhibition and connect them to SOX2 activity as well as transcription changes. Does activation of this SOX2-enhancer function (e.g., CRISPR activation) in NPC48h reverse the effect of DOT1L inhibition?

The effect of DOT1L inhibition on SOX2 enhancers is indeed a key outcome of our integrative analysis, which is now summarized in the new Figure 5. We have also followed the reviewers'

suggestion and included additional data for chromatin accessibility in mESC. This was a very helpful direction, as it highlights that the observed effects of DOT1L inhibition on SOX2 enhancers are specific to NPC. In mESC, we observe only small changes in both histone marks (Supplementary Figure 4c) and chromatin accessibility (Supplementary Figure 5a). Motif enrichment analysis reveals a distinct set of sequence features (Figure 5b), but no preferential binding of Sox2 on enhancers with reduced accessibility (Supplementary Figure 5d). Furthermore, we do not see any clear association between dynamic accessibility and gene expression changes induced by DOT1L inhibition (Supplementary Figure 5b).

Using SOX2 ChIP followed by qPCR, we have now validated that DOT1L inhibition results in a decrease of SOX2 binding at target enhancers in NPC48h (Supplementary Figure 5d). Hence our large-scale integrative analysis provides meaningful suggestions for further analysis, but a full mechanistic study on the effects of DOT1L inhibition on SOX2 enhancer activity is beyond the scope of this work.

Here are the suggestions for each main results:

FIRST PART:

1. Epigenetic and transcriptional changes during ESCs neuronal differentiation. Global profiling of seven histone modifications related to transcription states in ESCs and NPC48h showed H3K27ac loss and H3K79me2 gain. These two modification changes are not linked to each other. This conclusion should be clearly described in the result

compared with the previous finding (Godfrey et al.). This could strengthen the result from the RELACS based ChIP-seq assay.

From the *in-vitro* differentiation data, it is impossible to assess whether the global changes in H3K27ac and H3K79me2 are mechanistically linked. Upon DOT1L inhibition, on the other hand, H3K27ac appears to be independent from H3K79me2. Global quantification of histone modification changes upon EPZ treatment shows that decrease in H3K79me2 does not affect H3K27ac levels, nor the level of other histone modifications. This conclusion is now made more explicitly in the result section:

Lines 286-287: “Together, these data indicate that the genome-wide depletion of H3K79me2 did not result in a comparable global or local loss of H3K27ac.”

and in the discussion:

Lines 451-455: “We observe depletion of H3K27ac upon EPZ treatment, but this does not follow the global decrease in H3K79me2. Locally, however, loss of H3K27ac on enhancers and promoters alike correlates with transcriptional downregulation, and it is mirrored by a corresponding decrease in H3K4me3 on promoters. This data argues against the hypothesis that H3K79me2 is generally critical to preserve H3K27ac from being targeted by deacetylase complexes in our model ²⁴.”

It should be noted that the ChIP-seq generated by Godfrey et al. is not quantitative. This prevents the use of their dataset for global quantifications. On the other hand, they report a local loss of H3K27ac upon EPZ treatment on specific loci, which does not warrant a conclusion on the global cross-talk between these two histone modifications.

2. In Figure 2d/e, the authors focused on H3K79me2 and how this mark correlates with gene expression. Only genes in cluster 3 showed an increase in H3K79me2 downstream of TSS. These genes were also upregulated in NPCs compared to mESCs and have a neuronal development signature. The authors can elaborate more on this result. Is the accumulation of H3K79me2 causative for the upregulation of these genes or just a byproduct of their increased expression?

We believe this is a byproduct and we elaborate on this point in the discussion.

Lines 439-443: “Developmental gain of local enrichment of H3K79me2 associates with transcriptional activation of genes critical for neuronal differentiation, although the accumulation may be a byproduct of transcription. Previous reports [Chorey et al., 2019] have shown that the ectopic localisation of DOT1L to the promoter region of few target genes, and the consequent accumulation of H3K79me2, is not sufficient to initiate transcription.”

Chorey et al., <https://doi.org/10.1016/j.molcel.2018.10.028>

3. Is the expression of the genes in cluster 3 affected upon DOT1L inhibition? The authors mentioned this a bit in Figure 3b (cluster 3), but the message was not well delivered in the result, and the most relevant conclusion is in the discussion.

The set of transcriptionally deregulated genes upon DOT1L inhibition in NPC48h is not specifically associated with accumulation of H3K79me2 during differentiation (cluster 3, ϕ coefficient = 0.046). The same holds true for associations with any other cluster

identified in the clustering analysis presented in Figure 2d. For the reviewer, we have attached the corresponding heatmap below, showing the expression log2 fold changes induced by DOT1L inhibition, partitioned into the 5 clusters. We have also added a more explicit statement to the main text in the result section:

Lines 228-230: “Transcriptional deregulation in NPC48h was not associated with gain of H3K79me2 during differentiation (ϕ coefficient = 0.046, 65 DEG of 2170 genes gaining H3K79me2 during neuronal differentiation).”

4. H3K27ac and H3K79me2 show opposite trends during neuronal differentiation, with H3K27ac decreasing in NPCs compared to mESCs. In order “to explore candidate mechanisms accounting for these histone modification changes, the authors investigate the transcriptional dynamics of genes coding for epigenetic enzymes”. They conclude that since 4 out of 13 expressed HDACs show increased expression in NPCs, this could explain the decrease in H3K27ac levels. It would be interesting to use a proteomic study conducted in similar systems and test their findings.

The authors also described 2 HATs changes in NPCs. It seems Kat6a is mainly involved in K9ac, and Hat1 is reported to be involved in the acetylation of newly synthesized cytoplasmic histones. What is the exact connection between 2 HATs (Kat6a and Hat1) and H3K27ac decrease?

We agree with the reviewer that this section was very speculative.

The original analysis aimed at identifying all annotated genes coding for proteins with acetyltransferase or deacetylase functions, in the most unbiased way possible (meaning, regardless of their annotated specific substrates). In the first version of the manuscript, we reported the outcome of that analysis. We found no consistent trend in the expression fold change of acetyltransferases. We realize that the lack of a proteomics quantification on the actual abundance of HDAC in mESC and NPC48h limits our statement to a simple observation. For this reason, we have decided to remove this section, working towards a simplification of the results.

SECOND PART:

1. Epigenetic and transcriptional changes upon DOT1L inhibition. Most of all, the authors did not specify why they used a 10nM concentration of the inhibitor. Is it cited somewhere? Did the authors test different concentrations?

First, we would like to rectify the reported concentration for EPZ treatment. It was 10 μ M and not 10 nM. This was a typo, and we have corrected it in the M&M. The choice of 10 μ M is based on a previous work from Roidl et al., 2016 (<https://doi.org/10.1002/stem.2187>). Here, the authors test 3 concentrations, 1 μ M, 5 μ M and 10 μ M. The latter is the highest concentration that still allows for normal neuronal differentiation and best reproduces the effects of SGC0946 inhibition.

2. Figure 3a western blot seems to have differential loading of the H3 control protein for the NPC48h conditions. Is this accounted for in the WB quantification?

Yes, the quantification takes into account the intensity of H3. All computational steps taken in the densitometric analysis are available in: https://github.com/FrancescoFerrari88/code_DOT1L_paper/blob/master/figure_3/NOTEBOOKS/3a-3b_EPZ_vs_DMSO_H3K79me2_GLOBAL_BULCK_%26_LOCUS-SPECIFIC_CHANGES.ipynb

3. If DOT1L is the main focus of this study, which, as stated in the intro, regulates H3K79me_{1,2,3} - then immunoblots of H3K79me₁ and H3K79me₃ after EPZ treatment would be useful to show.

Thank you for pointing out this omission. We have now included an immunoblot where we show that EPZ treatment decreases all three degrees of methylation of H3K79. The result is now reported in Supplementary Figure 3a.

4. Considering the modest magnitude and numbers of the gene expression changes upon EPZ treatment in NPCs, is this enough to conclude that DOT1L inhibition mediates a shift in the transcriptional program of the progenitors? Are there any publicly available data showing a similar trend with DOT1L KD in NPCs?

We could not find any published dataset of DOT1L-KD in a similar system against which to compare our results.

Although the effects sizes and number of transcriptionally affected genes upon DOT1L inhibition are moderate in NPC48h, we think that the associated changes in epigenetic features (H3K4me₃, H3K27ac and chromatin accessibility) warrant the claim that DOT1L inhibition might bias the transcriptional state of neural progenitors. On this important point, we now explicitly elaborate in the discussion:

Lines 455-459: “In NPC48h, DOT1L inhibition seems to prime the transcriptome of neural progenitors towards neuronal differentiation. Although the transcriptional deregulation is moderate in effect size, epigenetic changes are associated with expression

dynamics, suggesting that the altered transcriptional state induced by DOT1L inactivation might be epigenetically memorized and preserved through mitotic divisions.”

THIRD PART:

1. Chromatin accessibility changes upon DOT1L inhibition and Sox2. PCA plot and hierarchical clustering for gene expression analysis upon EPZ treatment show separation only between replicates and not between treatments (SFig.3a). PCA plot for ATAC-seq data in NPCs shows an excellent separation between drug and vehicle treatment. The authors should clearly describe their results and add interpretation. Is it a technical issue with the experiment, or is there a biological explanation for that?

We believe there could be a biological explanation for that and we elaborate this further in the discussion (Lines 473-476). Briefly, during differentiation the enhancer landscape has been shown to be the most dynamic compared to any other epigenetic feature (Nord et al., 2013; <https://doi.org/10.1016/j.cell.2013.11.033>). Since inhibition of DOT1L primes NPC towards downstream differentiation (Franz et al. 2019; <https://doi.org/10.1093/nar/gky953>), we would expect to observe highest variance in the accessibility of the enhancer landscape. As ATAC-seq provides us with that information, the data allows for separation on the first principal component. We would also like to point out that the separation on PC1 is far from excellent. Only ~38% of the total variance is explained by PC1, while PC2 accounts for ~33%. This is barely sufficient to allow separation on PC1 based on treatment groups.

2. While the H3K79me2 level is correlated with differentiation, inhibition of its writing enzyme facilitates neuronal differentiation by reducing SOX2 accessibility to intronic, and intergenic enhancers. What is the function of the global H3K79me2 increase during differentiation? What is the relationship between H3K79me2 enrichment and SOX2 binding? Is it directly related to the H3K79 methylation level and/or a byproduct of the non-enzymatic effect of the DOT1L?

The relation between the global increase of H3K79me2 during differentiation and the effects of DOT1L inhibition in neural committed cells *in-vitro* remains unclear. With regard to the association between SOX2 binding and H3K79me2, we already show that enrichment of H3K79me2 is not always present in ATAC peaks that tend to lose accessibility upon EPZ treatment and are bound by SOX2. This observation suggests that the effects observed on SOX2 might be independent of the methyltransferase activity of DOT1L.

We now elaborate on these subjects in the discussion:

Lines 484-491: “Our data also indicate that H3K79me2 enrichment does not discriminate between dynamic and non-dynamic open regions in our system, suggesting that DOT1L inhibition may affect a subclass of enhancers independently from H3K79 methylation. While local H3K79me2 enrichment does not specifically associate with decreased accessibility of SOX2-bound enhancers, the relation between H3K79me2 global increase during differentiation and the effects of DOT1L inhibition on SOX2-enhancers remains obscure. Further studies should be conducted to elucidate whether the global

accumulation of H3K79me2 bears important consequences for the enhancer activity of neural committed cells.”

Reviewer #2 (Remarks to the Author):

In this tour-de-force manuscript, Ferrari et al perform a multi-omic analysis of epigenetic, transcriptional and chromatin changes that occur during neuronal differentiation from mESCs to NPCs. They demonstrate that two of the seven histone marks they analyze H3K27ac and H3K79me2 surprisingly change in opposite directions (with minimal change in the other histone marks), differentiate between global and local epigenetic changes, and examine the effect that inhibition of DOT1L in mESC and mESC-derived NPC lines has on these cells, showing that inhibition of DOT1L is sufficient to initiate neuronal differentiation.

This study and analysis are excellently executed, the findings are novel and contribute greatly to the field and I recommend it for publication in Nature Communications with the following non-experimental, minor modifications/revisions:

- 1) On line 23,, define “MLL” as it is the first time you are using the term.
- 2) The second half of the compound sentence ending on line 70 is not a sentence and needs a verb.
- 3) Line 73 change “to detect” to “detection of.”
- 4) Line 95, get rid of “e.g.” in the sentence.
- 5) The bulk of the RESULTS section needs to be changed to the past tense instead of the present tense, when explaining what the researchers did. Interpretation of the data can remain in the present tense, but the experiments and analysis need to be in the past tense.

- 6) Line 136, the outermost parenthesis need to be changed to brackets “[]” while the inner ones can remain curved.
- 7) Line 145, tell us which TSS and TTS size ranges were used for each modification (like you do on line 199) and why those regions were chosen, e.g. “since such and such histone mark is associated with proximal promoter regions...”, or “since such and such is associated with the ends of coding regions...”.
- 8) Line 193, the word “Eventually” is the wrong word and could be omitted completely or replaced with a word like “furthermore” depending on what exactly you wish to convey.
- 9) Line 258, same as above for “Eventually.”
- 10) Line 279, remove the “to” in the phrase “signal equals to 47.8%...”
- 11) Line 408, change “to dissect” to “the dissection of.”
- 12) Line 458, “enhancer” changed to “enhancers.”
- 13) Line 482, change it so it reads, “affected on ATAC-Down enhancers depending on H3K79me2 level (26), we visualized...”
- 14) Line 502, there is an extra “s” in the sentence.
- 15) Line 592, change it so it reads “In addition, here we present the first evidence...”

We would like to thank Dr. Johnson for his careful reviewing efforts and the positive assessment. We have included all his corrections and suggestions in the new manuscript.

Again, the experiments and analysis in this manuscript are impressive and well written, and I recommend publication in Nature Communications.

Steven M. Johnson. Ph.D.

Associate Professor

Assistant Chair

Department of Microbiology & Molecular Biology

Brigham Young University

REVIEWERS' COMMENTS:

Reviewer #1 (Remarks to the Author):

The authors have provided new data to the manuscript and addressed the reviewers' concerns. I appreciate that the authors have tried their best with analyzing RNA-seq and ATAC-seq after EPZ treatment using various bioinformatic tools. The analysis outcome is rather negative (minimum changes in gene expression upon global depletion of H3K79 methylation) and does not support the previously proposed model (no link between H3K79 methylation and H3K27 acetylation). However, these are important clarifications that the chromatin field will value.

Meanwhile, the authors' analysis also made two positive observations – gene expression changes associated with chromatin signatures and Sox2 binding decrease in a subset of regulatory elements upon global depletion of H3K79 methylation. These two observations are not connected, and the molecular mechanism of individual observation needs to be clarified in a further study.

Overall the revision manuscript would be suitable for publication in Nature Communications. However, I feel that the writing needs to be streamlined, in particular, the newly added part including abstract, result, and discussion. I also suggest putting the corresponding heatmap into the supplementary information.

Detailed response to Reviewers

Reviewer #1 (Remarks to the Author):

The authors have provided new data to the manuscript and addressed the reviewers' concerns. I appreciate that the authors have tried their best with analyzing RNA-seq and ATAC-seq after EPZ treatment using various bioinformatic tools. The analysis outcome is rather negative (minimum changes in gene expression upon global depletion of H3K79 methylation) and does not support the previously proposed model (no link between H3K79 methylation and H3K27 acetylation). However, these are important clarifications that the chromatin field will value.

Meanwhile, the authors' analysis also made two positive observations – gene expression changes associated with chromatin signatures and Sox2 binding decrease in a subset of regulatory elements upon global depletion of H3K79 methylation. These two observations are not connected, and the molecular mechanism of individual observation needs to be clarified in a further study.

Overall the revision manuscript would be suitable for publication in Nature Communications. However, I feel that the writing needs to be streamlined, in particular, the newly added part including abstract, result, and discussion. I also suggest putting the corresponding heatmap into the supplementary information.

We would like to thank the reviewer for the careful reading of our manuscript and the valuable suggestions. Together with editorial help, we have adjusted the Abstract and streamlined both Results and Discussion. We have now also included the heatmap from our previous correspondence with the reviewers, showing expression changes of protein coding genes in mESC and NPC48h upon EPZ treatment, for each of the five clusters presented in Figure 2d. This figure is now added as Supplementary Figure 4c. Please notice that the labeling of all subsequent supplementary figures was changed as a result.